# Reactive Oxygen Species Mechanisms that Regulate Protein–Protein Interactions in Cancer

**DOI:** 10.3390/ijms25179255

**Published:** 2024-08-27

**Authors:** Stavros Iliadis, Nikolaos A. Papanikolaou

**Affiliations:** Laboratory of Biological Chemistry, Department of Medicine, Section of Biological Sciences and Preventive Medicine, Aristotle University of Thessaloniki School of Medicine, 54124 Thessaloniki, Macedonia, Greece; siliad@auth.gr

**Keywords:** ROS, PPI, cancer, degradation, phosphorylation, kinases, phosphatase

## Abstract

Reactive oxygen species (ROS) are produced during cellular metabolism and in response to environmental stress. While low levels of ROS play essential physiological roles, excess ROS can damage cellular components, leading to cell death or transformation. ROS can also regulate protein interactions in cancer cells, thereby affecting processes such as cell growth, migration, and angiogenesis. Dysregulated interactions occur via various mechanisms, including amino acid modifications, conformational changes, and alterations in complex stability. Understanding ROS-mediated changes in protein interactions is crucial for targeted cancer therapies. In this review, we examine the role that ROS mechanisms in regulating pathways through protein–protein interactions.

## 1. Introduction

The regulation of protein–protein interactions serves as the fundamental framework that governs most cellular processes [1,2]. However, these interactions are not static; they are subject to the dynamic influence of reactive oxygen species (ROS), which are key players in redox biology (Figure 1) [3]. ROS, generated during normal cellular metabolism and in response to environmental stressors, have a multifaceted impact on the delicate balance of these interactions [4,5]. In this context, we explore the role of ROS in protein–protein interactions, from their modulation of protein conformations to their indirect effects on kinase and phosphatase activity, the stabilization of protein complexes through disulfide bond formation, and the regulation of protein degradation. The nuanced interplay between ROS and protein–protein interactions provides a window into the web of cellular signaling, offering insights into various physiological processes, disease mechanisms, and potential therapeutic interventions. This exploration draws upon recent research findings that have delved into the latest advancements in redox biology [6,7,8,9].

Protein–protein interactions are the backbone of almost all essential cellular processes [10,11]. These interactions dictate the behavior of signaling networks [12], the coordination of cellular responses [13,14], and the fine-tuning of regulatory mechanisms [15,16]. However, they are far from static. One of the most influential orchestrators of this biological symphony are reactive oxygen species (ROS) [17,18]. These chemically reactive molecules, arising as byproducts of cellular metabolism and environmental influences, can signal by inducing altered interactions: for example, they modulate tetradecanoyl-phorbol-13-acetate-induced interactions (TPA) of the Raf kinase inhibitory protein, causing its dissociation from the heat-shock protein 60/mitogen-activated protein kinase (ERK and JNK) complex [19,20].

The impact of ROS on protein–protein interactions is multifaceted. ROS can alter protein conformation [21,22], indirectly affecting the activity of crucial enzymes such as kinases [23] and phosphatases [24], stabilizing protein complexes through disulfide bond formation, and even regulating protein degradation pathways [25,26]. This complex interplay between ROS and protein–protein interactions serves as a critical nexus in the broader landscape of redox biology, offering insights into fundamental physiological processes, underlies mechanisms of disease, and has the potential for innovative therapeutic interventions [27,28]. Extensive research work has revealed how ROS, those double-edged molecular swords, shape the dynamics of protein–protein interactions and, in turn, the fate of cells and organisms. In this review, we examine the pathways and implicated mechanisms in different tumor types affected by ROS, and also, we explore some applications in health and disease. In the following paragraphs, we examine several important pathways and the mechanisms involved in modulating their PPIs in different types of tumors.

## 2. Pathways

### 2.1. The Ras-PI3K-Akt Pathway in Pancreatic Cancer

The Ras-PI3K-Akt pathway controls cellular growth and proliferation in several different cancer types, including pancreatic cancer [29,30]. ROS can disrupt this pathway by oxidizing specific cysteine residues in Ras and PI3K, which prevents their interaction and activation of downstream signaling. This leads to the inhibition of Akt signaling and subsequent cell death. Treatment with the ROS-inducing agent menadione causes a decrease in Akt phosphorylation and subsequent cell death in pancreatic cancer cells, which is mediated by the disruption of the Ras-PI3K-Akt pathway [31,32]. ROS-induced oxidation of the cysteine residue C118 in Ras prevents its interaction with PI3K and inhibits downstream Akt signaling [33]. Thus, ROS can target specific cysteine residues in Ras and PI3K proteins, which leads to disruption of their protein–protein interactions and subsequent inhibition of the normal PI3K-Akt signaling pathway. Similarly, in PI3K, cysteine residues in the p110α subunit, C862, are susceptible to ROS-induced oxidation. Oxidation of these cysteine residues disrupts the interaction of PI3K with Ras and other signaling molecules, leading to inhibition of the PI3K-Akt pathway [34,35,36,37,38,39,40]. 

Previous studies have highlighted the significance of ROS-induced oxidation in targeting specific cysteine residues within the Ras and PI3K proteins, effectively disrupting their protein–protein interactions and consequently suppressing the PI3K-Akt signaling pathway. ROS do not limit the effects on Ras and PI3K alone. They can also influence the activity of upstream signaling molecules, including receptor tyrosine kinases (RTKs) (Figure 2) [41]. These findings underscore the multifaceted role of ROS in influencing the Ras-PI3K-Akt pathway in pancreatic cancer, primarily through the oxidation of critical cysteine residues within key proteins. Such studies contribute significantly to our understanding of the molecular mechanisms involved in cancer development and provide potential avenues for targeted cancer therapy. Overall, ROS-induced oxidative modifications of cysteine residues on key signal transducers can disrupt protein–protein interactions critical for the activation of the Ras-PI3K-Akt pathway, which leads to the inhibition of downstream signaling and subsequent cell death [42].

### 2.2. Bcl-2-Family Proteins in Breast Cancer 

The Bcl-2 family of proteins plays a critical role in regulating apoptosis and the survival of cancer cells [44,45]. In breast cancer cells, ROS can disrupt protein–protein interactions between Bcl-2 and pro-apoptotic members of the Bcl-2 family, such as Bax and Bak [46]. This disruption leads to the activation of apoptosis and subsequent cell death. Treatment with the ROS-inducing agent paraquat causes an increase in the Bax/Bcl-2 ratio and subsequent apoptosis in breast cancer cells, which has been found to be mediated by the disruption of the Bcl-2/Bax interaction [47]. One example of this is the interaction between Bax and Bak, which are two pro-apoptotic members of the Bcl-2 family. ROS induce the formation of disulfide bonds between cysteine residues in Bax and Bak, enhancing their interaction and promoting the formation of apoptotic pores in the mitochondrial membrane. This ultimately leads to the release of cytochrome c and the activation of the caspase cascade, resulting in apoptosis [48]. The reversible formation of disulfide bonds between interacting proteins is targeted by ROS, and in the future, the identification of more instances may provide avenues for the targeted protection of such interactions. Disulfide bond formation, which has the potential to mediate extensive yet fully reversible structural and functional changes, rapidly adjusting the activity of the protein to prevailing oxidant levels, is a frequent target of ROS [45,46,47,48,49].

Another member of the Bcl-2 family of anti-apoptotic proteins targeted by ROS includes Mcl-1. ROS can oxidize cysteine residues in Bcl-2 and Mcl-1, leading to changes in their conformation and activity [50,51]. These changes can alter the balance between pro- and anti-apoptotic Bcl-2 family members and contribute to the development and progression of cancer. Overall, ROS-induced oxidative modifications can affect the interactions between members of the Bcl-2 family, leading to changes in apoptotic signaling and cell survival. Dysregulation of the Bcl-2 family is a hallmark of many types of cancer, and understanding the role of ROS in regulating these interactions could provide insights into the development of novel cancer therapeutics. In the wider context, these ROS-induced oxidative modifications significantly affect interactions within the Bcl-2 family. Delving into the role of ROS in this regulation presents an exciting avenue for the development of novel and targeted cancer therapeutics.

### 2.3. E-Cadherin and β-Catenin in Colorectal Cancer

ROS can also regulate the activity of the Wnt/β-catenin/E-cadherin pathway, which plays a critical role in cell proliferation, differentiation, and apoptosis [52,53]. E-cadherin is a critical cell adhesion molecule that plays a key role in maintaining epithelial tissue integrity. In colorectal cancer, elevated ROS levels can disrupt the protein–protein interactions between E-cadherin and β-catenin, leading to the activation of Wnt/β-catenin signaling and subsequent tumor growth [54]. Treatment with the ROS-inducing agent H_2_O_2_ causes a decrease in E-cadherin/β-catenin interactions and subsequent activation of Wnt/β-catenin signaling in colorectal cancer cells [55]. ROS can induce the oxidation of cysteine residues in E-cadherin, leadng to disruption of its interactions with β-catenin and subsequent destabilization of adherens junctions [56]. This results in increased cell motility and invasion, which can contribute to the metastasis of cancer cells. Additionally, ROS can oxidize the cysteine residues in β-catenin, which affects stability. This oxidation of β-catenin leads to its phosphorylation and subsequent degradation, resulting in decreased levels of β-catenin and subsequent changes in downstream signaling pathways [57,58]. ROS can induce the oxidation of specific cysteine residues in key components of this pathway, such as Dishevelled and Axin, which can lead to changes in downstream signaling and subsequent cancer growth [59,60]. Overall, ROS-induced oxidative modifications can affect the interactions between E-cadherin and β-catenin, leading to changes in cell adhesion and motility, which can contribute to cancer cell metastasis [61]. 

The introduction of the ROS-inducing agent H_2_O_2_ has been shown to decrease the interaction between E-cadherin and β-catenin, subsequently activating Wnt/β-catenin signaling in colorectal cancer cells [62]. In colorectal cancer, elevated H_2_O_2_-mediated ROS levels have been observed to disrupt protein–protein interactions between E-cadherin and β-catenin. This disruption has far-reaching consequences, leading to the activation of the Wnt/β-catenin signaling pathway, which, in turn, fuels tumor growth [63,64]. This study exemplifies the critical role of ROS in modulating these interactions within cancer contexts. The destabilization of E-cadherin/β-catenin interactions enhances cell motility and invasiveness, contributing to the metastatic potential of cancer cells [65]. In conclusion, the impact of ROS-induced oxidative modifications on the interactions between E-cadherin and β-catenin has profound implications for cell adhesion, motility, metastasis, and the development of colorectal cancer as well as other types of cancer.

### 2.4. HIF-1α and p300 

The transcription factor HIF-1α plays a critical role in regulating cellular responses to hypoxia in cancer cells. In glioblastoma cells, elevated ROS levels can disrupt the protein–protein interactions between HIF-1α and the transcriptional co-activator p300, which prevents the activation of HIF-1α target genes and subsequent tumor growth [66,67,68]. Menadione, a ROS-inducing agent, has been shown to decrease HIF-1α target gene expression and induce cell death in glioblastoma cells. The disruption of HIF-1α/p300 interactions is one proposed mechanism behind this effect. HIF-1α (Hypoxia-Inducible Factor 1-alpha) typically requires the coactivator p300 to drive the expression of its target genes under hypoxic conditions, which promote tumor survival. ROS generated by menadione can interfere with this interaction, leading to reduced gene expression and increased cell death in glioblastoma cells. Another study showed that ROS-induced oxidation of specific cysteine residues in p300 disrupted its interactions with HIF-1α and inhibited its transcriptional activity [69,70]. ROS can affect the activity of HIF-1α by inducing the oxidation of specific cysteine residues in its oxygen-dependent degradation domain (ODD), which stabilizes the protein and enhances its transcriptional activity. ROS can affect the activity of p300 by inducing the oxidation of specific cysteine residues in its histone acetyltransferase (HAT) domain, which affects its ability to acetylate histones and activate transcription. Additionally, ROS can also affect the interactions between p300 and other transcription factors, such as p53 and NF-κB, which can alter downstream signaling pathways and subsequent cancer growth. p300 is a multifunctional co-activator protein that interacts with a wide range of transcription factors to activate gene expression. One of the key transcription factors that interacts with p300 is p53, a tumor-suppressor protein that plays a critical role in preventing cancer by inducing cell-cycle arrest, DNA repair, and apoptosis in response to cellular stress. ROS lter the activity of p53 by inducing the oxidation of specific cysteine residues within its DNA-binding domain, which alters its ability to bind to DNA and activate transcription. The interaction between p300 and p53 is critical for the regulation of p53-mediated transcriptional activation. P300 acts as a co-activator of p53 by acetylating histones at the promoter regions of p53 target genes, which leads to increased expression of these genes. ROS-induced oxidation of p300 can affect its ability to acetylate histones and activate transcription, which in turn affects the ability of p53 to induce its target genes [71,72,73], finds its activity modulated by ROS-induced disruption in its protein–protein interactions with the transcriptional co-activator p300. This disturbance ultimately prevents the activation of HIF-1α target genes, thereby inhibiting tumor growth.

P53, a critical tumor-suppressor protein, plays a fundamental role in preventing cancer by inducing cell-cycle arrest, DNA repair, and apoptosis in response to cellular stress such as DNA damage. ROS, however, can modulate p53’s activity by oxidizing specific cysteine residues within its DNA-binding domain. This oxidative event alters p53’s DNA-binding capacity and its ability to activate transcription. Concurrently, ROS-induced oxidation of p300 can also affect its capacity to acetylate histones and activate transcription [74,75]. The interplay between p300 and p53 holds particular significance in the regulation of p53-mediated transcriptional activation. P300 acts as a co-activator for p53 by acetylating histones at the promoter regions of p53 target genes, resulting in increased gene expression. The impact of ROS-induced oxidation of p300 extends to this critical interaction, affecting p300’s ability to acetylate histones and, consequently, influencing p53’s capacity to induce its target genes. Dysregulation of the p300/p53 pathway has been implicated in the development and progression of various cancer types. Delving into the role of ROS in regulating these interactions provides valuable insights into the mechanisms governing cancer growth and opens new avenues for potential therapeutic targets in the battle against cancer.

### 2.5. Effects of ROS on Protein–Protein Interactions in Pathways of Brain and Lung Cancers

Reactive oxygen species (ROS) are critical regulators of protein–protein interactions (PPIs) within the signaling pathways of brain and lung cancers. These interactions significantly influence cancer progression, cancer metastasis, and cancer patients’ responses to therapy. In brain and lung cancers, ROS can modify proteins through redox reactions, which particularly affect cysteine residues. These modifications can alter the structure and function of proteins, impacting PPIs that are essential for signaling pathways involved in cancer progression. For example, ROS can modify the KEAP1 protein, leading to the stabilization of NRF2, which is crucial for cellular defense mechanisms against oxidative stress in lung cancer [76,77]. ROS influences several signaling pathways by modulating PPIs. In brain cancers, as in other types of tumors, ROS can affect interactions within the PI3K-Akt pathway, which is vital for cell survival and proliferation. In lung cancer, ROS can modify PPIs involved in the epithelial–mesenchymal transition (EMT), a process that facilitates metastasis. By altering interactions between proteins such as E-cadherin and vimentin, ROS promote the invasive capabilities of cancer cells [78]. Moreover, ROS can impact PPIs in pathways related to angiogenesis and invasion in brain cancers. For instance, ROS can modulate interactions between vascular endothelial growth factor (VEGF) and its receptors, promoting angiogenesis and contributing to the aggressive nature of these tumors [79]. ROS acts as signaling molecules that can either promote or inhibit cancer progression depending on their concentration. At moderate levels, ROS activate signaling cascades involving mitogen-activated protein kinase (MAPK), extracellular signal-regulated kinase (ERK), and nuclear factor kappa-light-chain enhancer of activated B cells (NF-κB), which promote cancer cell survival and proliferation. At higher concentrations, ROS can induce apoptosis through intrinsic and extrinsic pathways, involving mitochondrial damage and death receptor activation, respectively [77,80]. 

### 2.6. Interactions between Reactive Oxygen Species (ROS) and Lipids: Implications for Protein–Protein Interactions

Reactive oxygen species (ROS) are pivotal in the oxidation of lipids, a process known as lipid peroxidation, which significantly influences protein–protein interactions (PPIs) in cells. This process is especially relevant in the context of cancer, where lipid peroxidation products can alter signaling pathways and contribute to oncogenic processes. Lipid peroxidation is initiated when ROS, such as hydroxyl radicals, attack polyunsaturated fatty acids (PUFAs) in cell membranes. This reaction results in the formation of lipid radicals, which propagate chain reactions, leading to the production of lipid peroxides. These lipid peroxides can decompose into reactive aldehydes, such as 4-hydroxynonenal (4-HNE), malondialdehyde (MDA), and acrolein, which are capable of forming adducts with proteins, nucleic acids, and other cellular components [81,82]. Among the lipid peroxidation products, 4-HNE is particularly significant due to its high reactivity. 4-HNE can form covalent adducts with cysteine, histidine, and lysine residues on proteins, thereby altering their structure and function. This modification can disrupt PPIs and affect signaling pathways involved in cell proliferation, apoptosis, and stress responses. For example, 4-HNE can modify proteins involved in the NF-κB signaling pathway, leading to altered transcriptional activity and promoting oncogenic signaling [83]. 

The formation of 4-HNE–protein adducts can have profound effects on PPIs. These modifications can either inhibit or enhance protein interactions, depending on the specific proteins and pathways involved. For instance, in cancer cells, 4-HNE has been shown to modify key signaling proteins, thereby influencing pathways such as MAPK and PI3K-Akt, which are crucial for cell survival and proliferation. The role of lipid peroxidation in modulating PPIs and signaling pathways highlights its potential as a therapeutic target. By understanding the specific modifications induced by lipid peroxidation products like 4-HNE, novel therapeutic strategies can be developed to inhibit these interactions and restore normal signaling. Antioxidants and inhibitors of lipid peroxidation are being explored as potential therapies to mitigate the effects of ROS-induced lipid damage in cancer. The formation of reactive aldehydes such as 4-HNE and their ability to form adducts with proteins underscore the complexity of ROS interactions in cancer biology. Understanding these mechanisms offers valuable insights for developing targeted therapies aimed at disrupting these pathological interactions.

To summarize, 4-hydroxynonenal (4-HNE) is a highly reactive aldehyde formed during lipid peroxidation, and it plays a significant role in altering protein–protein interactions (PPIs) through its ability to form covalent adducts with proteins. These modifications can profoundly affect cellular signaling pathways and contribute to oncogenic processes. 

### 2.7. Mechanisms of 4-HNE’s Interactions with Proteins 

4-HNE primarily interacts with proteins by forming covalent adducts with nucleophilic amino acid residues such as cysteine, histidine, lysine, and arginine. This interaction can occur through Michael addition or the formation of Schiff bases, leading to changes in protein structure and function [84,85]. For instance, 4-HNE can modify the catalytic sites of enzymes or bind to residues outside the active site, thereby altering enzyme activity and protein interactions [86]. As discussed above, the formation of 4-HNE–protein adducts can disrupt PPIs by altering the structural conformation of proteins. This disruption can lead to changes in signaling pathways that regulate cell proliferation, apoptosis, and stress responses. For example, 4-HNE can modify proteins involved in the NF-κB signaling pathway, leading to altered transcriptional activity and promoting inflammation and oncogenic signaling. In addition to its role in signaling pathways, 4-HNE can influence the dynamics of PPIs by modifying proteins such as tubulins, which are essential for microtubule formation. 4-HNE’s adduction to tubulin can impair its polymerization, leading to microtubule disorganization and affecting cellular processes such as mitosis and intracellular transport. 

### 2.8. Biological Implications and Oncogenic Potential of 4-HNE

The ability of 4-HNE to form adducts with proteins makes it a potent modulator of cellular processes. These modifications can lead to the activation of stress-response pathways and contribute to the development and progression of cancer. For instance, 4-HNE has been shown to modulate ligand-independent signaling via membrane receptors such as EGFR, acting as a sensor of external stress and initiating signaling cascades that affect cell growth and survival [87]. Given the significant impact of 4-HNE on PPIs and signaling pathways, targeting 4-HNE–protein interactions is a promising therapeutic strategy. Understanding the specific proteins and pathways affected by 4-HNE can lead to the development of interventions aimed at mitigating its oncogenic effects. Antioxidants and compounds that can detoxify 4-HNE or prevent its formation are being explored as potential therapies to counteract the deleterious effects of lipid peroxidation products. In summary, 4-HNE plays a crucial role in altering protein–protein interactions, thereby influencing cellular signaling and contributing to oncogenic processes. Its ability to modify proteins and disrupt PPIs highlights the complexity of its role in cancer biology and underscores the potential for therapeutic interventions targeting 4-HNE-mediated modifications.

### 2.9. Therapeutic Implications 

Understanding the specific PPIs affected by ROS in brain and lung cancers provides opportunities for therapeutic interventions. Targeting these interactions can disrupt critical signaling pathways, potentially inhibiting cancer progression. For example, therapies that modulate ROS levels can either increase ROS to induce cancer cell death or decrease ROS to prevent tumor progression, depending on the cancer type and stage. Antioxidants may be used to reduce ROS levels, thereby inhibiting pro-tumorigenic signaling pathways. However, caution is necessary when utilizing antioxidants as they can also protect cancer cells from ROS-induced apoptosis. Alternatively, therapies that specifically target ROS-sensitive PPIs could provide a more effective approach, especially when combined with other treatments that manipulate ROS levels. In summary, ROS significantly impact protein–protein interactions in pathways critical to brain and lung cancers. By understanding these interactions, researchers can develop targeted therapies that exploit the dual nature of ROS, offering potential advancements in cancer treatment [78,80]. 

## 3. Mechanisms

### 3.1. Direct ROS-Mediated Mechanisms

Reactive oxygen species (ROS) can modulate protein complex interactions in cancer cells through multiple mechanisms [28]. ROS can affect protein conformation: they oxidize amino acid residues in proteins, leading to changes in protein structure and function. These changes can alter the conformation of proteins that are part of a protein complex, and, as a consequence, alter the interactions between these proteins [35,88]. The first mechanism in this process involves changes in protein conformation. For example, ROS-mediated oxidation of the transcription factor NF-κB can lead to a conformational change that disrupts its interaction with its inhibitor, IκB, thereby activating NF-κB target genes [89]. In its inactive state, NF-κB is held in check by its inhibitor, IκB, which prevents NF-κB from translocating to the cell nucleus where it would initiate gene transcription. However, the introduction of ROS can lead to the oxidation of specific amino acid residues within NF-κB. While cysteine is the primary target for oxidation in NF-κB and its regulatory proteins, it is worth noting that ROS can potentially oxidize other amino acids, such as arginine, lysine, proline, threonine, tryptophan, phenylalanine, and histidine. Also, ROS can directly oxidize IκB at cysteine sites, inducing the nuclear translocation of NF-κB. This oxidative event initiates a conformational change in the transcription factor, disrupting its interaction with IκB. On the other hand, the IKK complex, which regulates NF-κB activation, is also a target for ROS-mediated oxidation. Specifically, H_2_O_2_ can inactivate IKK by oxidizing cysteine 179 of IKKβ, leading to S-glutathionylation and reduced NF-κB signaling. Conversely, H_2_O_2_ can activate IKK in some cell types by forming disulfide bonds between Cys54 and Cys347 of the IKKγ/NEMO subunit.

This disruption frees NF-κB from its inhibitory partner, allowing it to translocate into the nucleus and activate its target genes, which are involved in a plethora of cellular responses, making NF-κB a central regulator of processes like inflammation and immune reactions [90]. Reactive oxygen species (ROS), such as superoxide, hydrogen peroxide, and hydroxyl radicals, have the remarkable ability to influence the conformation and activity of proteins within cells [91], with several proteins acting as redox sensors. This dynamic interplay between ROS and protein conformation hinges on the oxidative modification of specific amino acid residues within these proteins, ultimately culminating in structural changes that can profoundly impact their functions [92]. The NF-κB/IκB interaction exemplifies this interplay between ROS and protein conformation, underscoring the role of ROS as signaling molecules that can alter vital cellular responses. 

Another notable example is the FOXO transcription factor family, comprising important regulators of cellular stress responses and cellular antioxidant defense [93]. ROS, as well as other stressful stimuli that elicit the formation of ROS, regulate this family of proteins by modulating FoxO activity at multiple levels, including post-translational modifications of FOXOs (such as phosphorylation and acetylation) [94], affecting their interactions with co-regulators [95], and inducing alterations in the subcellular localization of FoxO [96], protein synthesis, and stability. ROS-induced post-translational modifications play a significant role in this regulation, including phosphorylation by activated c-Jun N-terminal kinase (JNK), which promotes FOXO nuclear localization and transcriptional activity. Oxidative stress also leads to the acetylation of FOXO proteins via the activation of p300/CREB-binding protein, and can induce monoubiquitination, affecting FOXO activity. FOXO proteins engage in redox-sensitive interactions with co-regulators and antioxidant peroxiredoxins. Increased intracellular ROS levels facilitate FOXO’s localization to the nucleus, where it is transcriptionally active. ROS can also influence FOXO protein stability, and there is evidence suggesting their redox-sensitive regulation of FOXO gene expression. While the referenced studies do not specify the exact amino acids modified in FOXO proteins by ROS, cysteine residues are often targets for oxidation due to their reactive thiol groups. The specific amino acids modified and the exact mechanisms can vary depending on the particular FOXO protein (FOXO1, FOXO3a, FOXO4, or FOXO6) and the cellular context. These diverse mechanisms collectively contribute to the complex regulation of FOXO transcription factors by ROS, influencing their activity and function in cellular processes [93].

### 3.2. Indirect ROS-Mediated Mechanisms

Indirect mechanisms involve ROS affecting protein–protein interactions through the modulation of protein kinases and phosphatases. Changes in protein phosphorylation can affect protein conformation and alter protein–protein interactions. For example, ROS-mediated activation of the protein kinase JNK can lead to phosphorylation of the transcription factor c-Jun, altering its conformation and enhancing its interaction with other proteins such as ATF-6, p38, and ERKs. The phosphorylation of c-Jun by JNK creates a sophisticated regulatory mechanism without inducing large-scale conformational changes. As an intrinsically disordered protein, c-Jun’s regulation occurs through a precise temporal pattern of phosphorylation at four specific residues within its transactivation domain. This sequential modification generates three distinct functional states of c-Jun, each with unique interaction profiles and cellular roles. The unphosphorylated state recruits the MBD3 repressor, while the doubly phosphorylated state (at serines 63 and 73) favors binding to the TCF4 co-activator. The fully phosphorylated state, involving all four sites, attenuates JNK signaling by disfavoring TCF4 binding [97]. These phosphorylation-dependent states modulate c-Jun’s ability to form complexes with various partners, including components of the AP-1 transcription factor. Additionally, phosphorylation enhances c-Jun’s stability by reducing its susceptibility to proteasomal degradation. This intricate system allows c-Jun to encode multiple functional states in response to JNK activation, enabling a nuanced and complex signaling response from a single kinase input [98,99].

Kinases and phosphatases play a pivotal role in protein phosphorylation, a post-translational modification, significantly influencing protein conformation and, consequently, protein–protein interactions [100,101]. This reversible modification serves as a molecular switch regulating various cellular processes. Alterations in the phosphorylation state of a protein can induce changes in its conformation, thereby affecting its interactions with other proteins. One illustrative example of ROS-mediated modulation through kinases involves the protein kinase JNK. ROS can stimulate JNK activity, leading to the phosphorylation of specific target proteins. In particular, JNK activation can result in the phosphorylation of the transcription factor c-Jun. This phosphorylation event causes a conformational change in c-Jun, rendering it more receptive to interactions with other proteins. In this case, c-Jun’s altered conformation enhances its affinity for another transcription factor, ATF-2 (Activating Transcription Factor 2). The strengthened interaction between c-Jun and ATF-2 can subsequently influence gene expression patterns, impacting cellular responses. In essence, ROS indirectly affect protein–protein interactions by triggering a cascade of events that ultimately lead to changes in protein phosphorylation. The ramifications of these modifications can extend beyond individual proteins and influence the complex network of interactions within the cell. The mechanisms of c-Jun phosphorylation and its effects on protein interactions are more nuanced than a simple conformational change. JNK phosphorylates c-Jun at four specific residues within its transactivation domain (TAD) in a precise temporal pattern. Serine 63 and serine 73 are phosphorylated more rapidly than threonine 91 and threonine 93. This sequential phosphorylation creates three distinct functional states of c-Jun, each with unique interaction profiles. The unphosphorylated c-Jun recruits the MBD3 repressor, maintaining a transcriptionally repressed state. When serine 63 and serine 73 are phosphorylated, c-Jun transitions to an active state wherein the MBD3 repressor is displaced, allowing the binding of the transcriptional activator TCF4. This doubly phosphorylated state triggers c-Jun’s transcriptional activity. Subsequently, the slower phosphorylation of threonine 91 and threonine 93 leads to a fully phosphorylated state that disfavors TCF4 binding, attenuating JNK signaling and returning c-Jun to a transcriptionally inactive state. These phosphorylation-dependent states modulate c-Jun’s ability to form complexes with various partners, including components of the AP-1 transcription factor such as JunB, JunD, c-Fos, and ATF. Additionally, phosphorylation enhances c-Jun’s stability by reducing its susceptibility to proteasomal degradation, leading to its accumulation. This intricate system allows c-Jun to encode multiple functional states in response to JNK activation, enabling a nuanced and complex signaling response from a single kinase input. It is important to note that c-Jun is an intrinsically disordered protein, and the phosphorylation events do not induce large-scale conformational changes in the traditional sense; instead, they create local changes in charge and structure that significantly alter c-Jun’s functional interactions and cellular behavior. This mechanism provides a sophisticated regulatory system that fine-tunes c-Jun’s activity in response to ROS-mediated JNK activation, allowing for the precise control of cellular responses to oxidative stress.

### 3.3. ROS-Mediated Protein Complex Stability

Another mechanism involves protein complex stability: ROS can affect protein complex stability by oxidizing cysteine residues in proteins, forming disulfide bonds that stabilize them. For instance, the formation of disulfide bonds between subunits of the transcription factor AP-1 can enhance its stability and promote the activation of target genes. Reactive oxygen species (ROS) play a crucial role in regulating the stability and activity of protein complexes, particularly transcription factors like AP-1, through the oxidation of cysteine residues. This mechanism is exemplified in the redox-sensitive regulation of the AP-1 complex. AP-1 is a dimeric transcription factor composed of proteins from the Jun, Fos, and ATF families. The c-Jun subunit of AP-1 contains three cysteine residues (Cys-269, Cys-320, and Cys-325) in its DNA-binding domain that are susceptible to oxidation. Under oxidative conditions, these cysteines can form intramolecular or intermolecular disulfide bonds, with Cys-269 being capable of bonding with either Cys-320 or Cys-325. This oxidation-induced disulfide bond formation significantly enhances c-Jun’s DNA-binding activity by increasing its affinity for target DNA sequences. Moreover, oxidation promotes the formation of c-Jun homodimers or heterodimers with other AP-1 components like c-Fos, further stabilizing the AP-1 complex. The c-Fos subunit also contains a redox-sensitive region in its DNA-binding domain, where cysteine oxidation enhances its DNA-binding activity and its ability to form stable dimers with c-Jun. Importantly, this process is reversible, allowing for the dynamic regulation of AP-1 activity in response to changing cellular redox conditions. Reducing agents can break these disulfide bonds, potentially decreasing AP-1 activity. This redox regulation mechanism provides cells with a rapid and reversible means to respond to oxidative stress, enabling quick activation of AP-1-dependent genes involved in various cellular processes, including stress responses, proliferation, and apoptosis. However, it is crucial to note that while oxidation can enhance AP-1 activity, excessive oxidation may lead to protein dysfunction, underscoring the importance of maintaining a delicate balance between oxidation and reduction for optimal AP-1 function and cellular homeostasis [102,103].

### 3.4. ROS-Mediated Protein Degradation

A third mechanism involves protein degradation. Reactive oxygen species (ROS) impact the regulation of protein degradation [104]. One key player in cellular protein degradation is the proteasome, responsible for breaking down intracellular proteins, some of which are components of essential protein complexes. ROS-mediated modulation of proteasome activity can lead to alterations in the stability and functionality of these complexes, presenting a critical aspect of redox biology [105]. The proteasome’s primary role is to maintain cellular homeostasis by degrading proteins tagged for disposal. This process involves the recognition and subsequent degradation of proteins, including those participating in various protein complexes. ROS can affect proteasome activity through various mechanisms, including direct oxidative modifications. When a proteasome’s activity is perturbed, changes in protein turnover can impact complex stability and function. Research is ongoing in the field of redox biology to explore ROS-mediated proteasome modulation, encompassing the specific mechanisms involved, downstream consequences, and implications for various cellular processes and disease states. Overall, ROS can modulate protein complex interactions in cancer cells through multiple mechanisms, including changes in protein conformation, protein–protein interactions, protein complex stability, and protein degradation. These changes can alter cellular signaling and gene expression, contributing to cancer development and progression.

## 4. Therapeutic Approaches Targeting ROS Levels and Protein–Protein Interactions in Cancer

### 4.1. Current Use of Antioxidant Medications and Supplements in Cancer Therapy

Antioxidants have long been considered as potential adjuncts in cancer therapy due to their ability to neutralize reactive oxygen species (ROS) and protect cells from oxidative stress. Common antioxidants include vitamin C, vitamin E, selenium, and glutathione, which are either administered as dietary supplements or as part of therapeutic regimens. The rationale behind using antioxidants in cancer therapy lies in their potential to reduce the oxidative damage caused by elevated ROS levels, which represent a hallmark of many cancers. However, the use of antioxidants in cancer treatment is complex and controversial. On one hand, antioxidants might protect normal cells from the oxidative damage induced by chemotherapy and radiation therapy, reducing side effects. On the other hand, they could also protect cancer cells, potentially reducing the efficacy of ROS-generating treatments and promoting tumor survival. Some studies suggest that antioxidants might interfere with the prooxidant effects of certain chemotherapies, which rely on ROS to kill cancer cells, thereby leading to reduced treatment efficacy [106].

Moreover, recent research has shown that ROS also play a role in disrupting protein–protein interactions (PPIs) within cancer cells. By oxidizing specific amino acid residues, ROS can alter the structure and function of proteins, leading to the formation or disruption of protein complexes that are crucial for cancer cells’ survival and proliferation. Therefore, while antioxidants might protect normal tissues, they could also inadvertently stabilize harmful PPIs in cancer cells, further complicating their therapeutic use [107].

### 4.2. Variation in ROS Levels across Different Stages of Cancer

ROS levels are not static and can vary significantly across different stages of cancer development and progression. In early stages, cancer cells often exhibit moderate increases in ROS, which can drive mutations and promote oncogenic signaling pathways. As a tumor progresses, metabolic reprogramming and mitochondrial dysfunction lead to further increases in ROS levels, contributing to more aggressive phenotypes and increased resistance to therapy. In advanced stages, cancer cells often adapt by upregulating antioxidant defenses in order to survive in the high-ROS environment. This adaptation is critical for maintaining the balance between ROS-driven proliferation and avoiding ROS-induced cell death. The variation in ROS levels across cancer stages highlights the potential for stage-specific therapeutic strategies. For example, in early-stage cancers, enhancing ROS levels might promote cancer cell death, whereas in late-stage cancers, targeting the antioxidant systems that allow cancer cells to thrive in high-ROS conditions might be more effective. These fluctuations in ROS levels also affect PPIs. In early stages, ROS-induced modifications might disrupt critical tumor-suppressing interactions, while in advanced stages, the stabilization of oncogenic PPIs by ROS could promote cancer progression. This dynamic suggests that targeting the ROS–PPI axis could be a promising therapeutic strategy, with different approaches being needed depending on the stage of the cancer [108,109].

### 4.3. The Role of Genetic Modifiers in ROS Sensitivity and the Development of ROS-Targeting Therapeutic Agents

Genetic factors play a crucial role in determining a cancer cell’s sensitivity to ROS. Variations in genes involved in ROS production, ROS detoxification, and repairing ROS-induced damage can significantly influence how cancer cells respond to oxidative stress. For instance, mutations in genes like TP53, which encodes the tumor-suppressor protein p53, can impair the cell’s ability to respond to ROS-induced DNA damage, leading to increased mutation rates and cancer progression [110]. The current understanding of genetic modifiers has paved the way for the development of ROS-targeting therapeutic agents. These agents aim to exploit the differential sensitivity to ROS between normal and cancer cells. For example, drugs that inhibit antioxidant defenses in cancer cells can selectively increase ROS to lethal levels, while sparing normal cells that have lower ROS levels. Additionally, targeting specific PPIs that are stabilized by ROS in cancer cells offers another avenue for therapy. By disrupting these PPIs, it may be possible to selectively kill cancer cells without harming normal tissue. These approaches highlight the potential of integrating genetic information with ROS-targeting strategies to develop personalized cancer therapies. By understanding the genetic landscape of a tumor, clinicians can better predict how it will respond to ROS modulation, thereby improving the efficacy of treatments [111,112,113].

## 5. Conclusions

The balance between the generation and elimination of reactive oxygen species (ROS) represents a critical determinant of cell fate and function. While ROS are vital participants in various physiological processes, their excessive accumulation can lead to detrimental damage to cellular components, ultimately resulting in cell death or malignant transformation. Importantly, ROS-mediated alterations in protein interactions have emerged as central players in the progression of cancer, impacting key cellular processes such as proliferation, migration, and angiogenesis. These perturbed protein interactions occur through multiple mechanisms, including modifications of amino acid residues, such as cysteine; conformational changes; and disruptions to the stability of complexes. A deeper understanding of how ROS affect protein interactions provides valuable insights into the web of signaling pathways that regulate cancer progression. Recognizing the critical role of ROS in modulating these pathways and their underlying mechanisms is of paramount importance for the development of targeted cancer therapies. This review has endeavored to shed light on the multifaceted role of reactive oxygen species in shaping cellular pathways and interactions, with a particular focus on their implications in cancer. Further dissecting the interplay between ROS and protein interactions will contribute to the ongoing efforts to harness this knowledge for the development of innovative and precise therapeutic strategies in the fight against cancer.

## Figures and Tables

**Figure 1 ijms-25-09255-f001:**
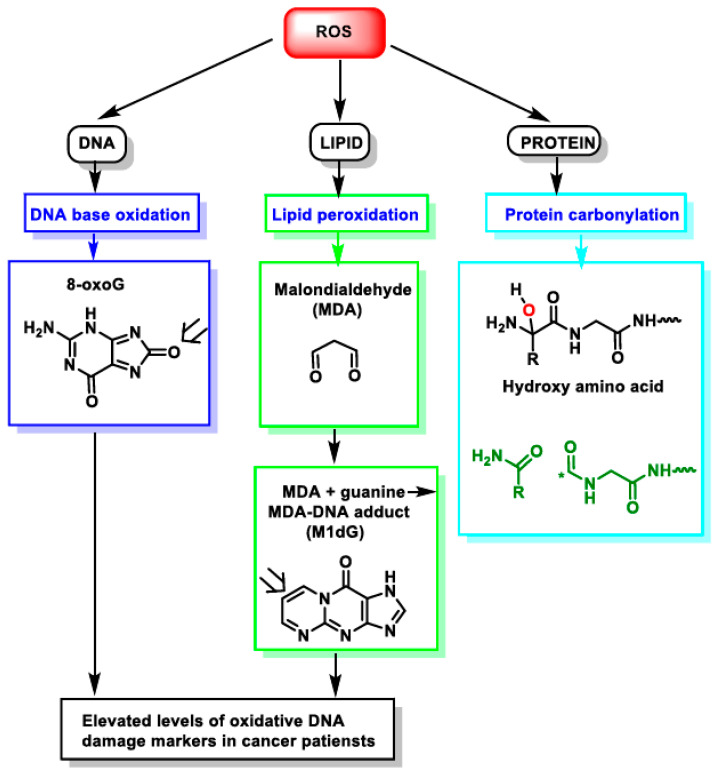
Impact of ROS on different macromolecules: unchecked ROS causes DNA base oxidation, lipid peroxidation, and protein carbonylation. *—unpaired electron.

**Figure 2 ijms-25-09255-f002:**
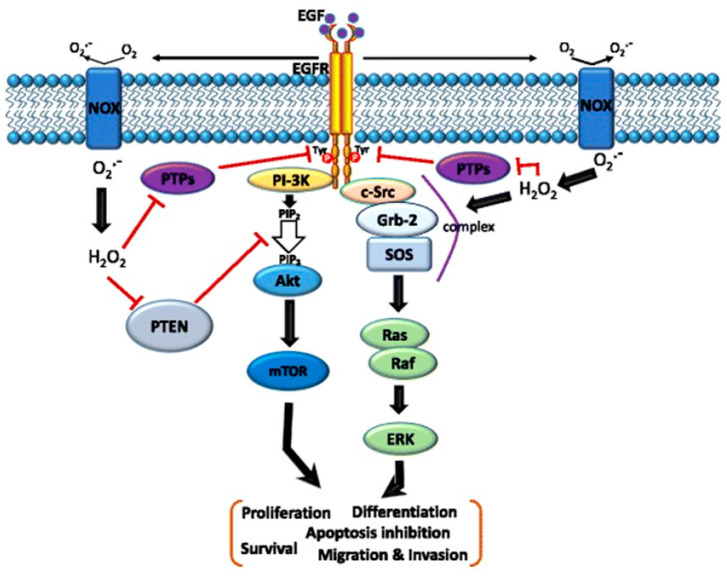
Schematic of the cross-talk between the epidermal growth factor (EGF)–EGF receptor (EGFR) axis and NADPH oxidase (NOX)-mediated reactive oxygen species (ROS) signaling pathways. The binding of EGF to EGFR induces receptor dimerization and then autophosphorylation of tyrosine (Tyr) residues (red circles) in its cytoplasmic domain. These phosphorylated Tyr residues serve as docking sites for associated proteins that activate multiple pathways. In particular, the Ras/Raf/mitogen-activated protein kinase (MAPK) and phosphatidylinositol-3-kinase (PI3K)/Akt pathways downstream of the EGFR play critical roles in cell migration, invasion, proliferation, and survival. Moreover, the EGF–EGFR axis also induces NOX-mediated hydrogen peroxide production, and hydrogen peroxide can diffuse across the membrane to reach the intracellular cytosol. Transient increases in hydrogen peroxide induce the oxidation of reduction–oxidation reaction (redox) targets such as phosphatase and tensin homolog (PTEN) to promote Akt activation, protein tyrosine (Tyr) phosphatases (PTPs) to enhance EGFR Tyr phosphorylation, or complex formation of SHC-Grb2-SOS with EGFR to activate Ras/MAPK signaling. Grb-2—growth factor receptor-bound protein 2; SOS—guanine nucleotide exchange protein [43].

## Data Availability

No new data were generated for this study.

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
