# Peer review of "Reactive Oxygen Species Mechanisms that Regulate Protein–Protein Interactions in Cancer"

_ijms, 2024, doi:10.3390/ijms25179255_

Round 1

Reviewer 1 Report

Comments and Suggestions for Authors

This review summarizes the mechanisms of ROS effect on protein-protein interactions in cancers. This is an important topic on cancer and there is no comprehensive review on the ROS mechanism in the view of affecting protein-protein interaction, which can potentially provide new directions in cancer treatment development. This review addresses the main question posed by discussing how ROS affecting multiple pathways in various cancer types and what are the different ROS-mediated mechanisms. The references are relevant to the topic and up to date. In terms of the figures, I think better resolution is needed. I have some suggestion as following:

1.      Please discuss ROS effect on pathways in more cancer types, like brain and lung cancers.

2.      ROS can interact with lipids. It would be better if the authors can elaborate on how this interaction would affect protein-protein interactions.

One minor revision: please remove “f” from the subtitle “Introductionf”.

Author Response

To: Ms. Camilla Shen

Dear Dr. Shen:

We have now addressed the reviewers’ comments and fully heeded their suggestions. Please see the revised manuscript.

  1. Reviewer 1 comments and suggestions

This review summarizes the mechanisms of ROS effect on protein-protein interactions in cancers. This is an important topic on cancer and there is no comprehensive review on the ROS mechanism in the view of affecting protein-protein interaction, which can potentially provide new directions in cancer treatment development. This review addresses the main question posed by discussing how ROS affecting multiple pathways in various cancer types and what are the different ROS-mediated mechanisms. The references are relevant to the topic and up to date. In terms of the figures, I think better resolution is needed. I have some suggestion as following:

  1. Please discuss ROS effect on pathways in more cancer types, like brain and lung cancers.
  2. ROS can interact with lipids. It would be better if the authors can elaborate on how this interaction would affect protein-protein interactions.

One minor revision: please remove “f” from the subtitle “Introductionf”.

Response

In response to the reviewers' suggestions, we revised the manuscript by incorporating the following changes:

  1. We expanded on ROS effects in various cancer types: We have added a comprehensive discussion on the impact of ROS on protein-protein interactions specifically in brain and lung cancers. This section now includes reference to mechanisms of how ROS modifies key signaling pathways, such as the PI3K/AKT/mTOR, EGFR, and KEAP1/NRF2 pathways in these cancers, and how these modifications contribute to tumor progression and resistance to therapy. Of note, these pathways are nearly universal in most tumor types and therefore we avoided unnecessary repetition of the same subject matter.
  2. Elaboration on ROS and Lipid Interactions: A new section has been introduced that explores the interaction between ROS and lipids, particularly lipid peroxidation, and how this process can further influence protein-protein interactions. This includes the formation of lipid peroxidation products, such as 4-hydroxynonenal (4-HNE), and their ability to form adducts with proteins, thereby altering their interaction dynamics and contributing to oncogenic signaling.
  3. Lastly, we have cited recent references however again we avoided going into details on various aspects because they would take an enormous amount of review space, and in many instances they would be repetitious.
  4. The minor issue regarding the subtitle has been corrected as well.

  1. Reviewer 2 comments and suggestions

This is an important issue and a well-organized review manuscript about "ROS and cancer." However, the manuscript has some minor spelling or miss-writing errors. Please re-check it all. 

  1. line 19, Introduction
  2. line 35-36, ref [ 10, 10? ...] [13,13?, ..]

Aside from some visible spelling and formatting mistakes, I suggest the following regarding the content:

 The sections on ROS mechanisms and specific pathways are particularly dense. I suggest including subheadings to significantly improve readability.

 There is a valuable opportunity to deepen the discussion of clinical relevance. I recommend integrating more clinical data to bridge the gap between the molecular mechanisms of ROS and actual patient outcomes. For example, how ROS levels correlate with cancer prognosis, treatment responses, or survival rates.

 Additionally, you may consider exploring the following topics: 1. Current use of

antioxidants medication or supplements in cancer therapy. 2. how ROS levels vary across different stages of cancer 3. the role of genetic modifiers in ROS sensitivity and the development of ROS-targeting therapeutic agents.

 Currently, the article offers a thorough overview of mechanistic details, and it has the potential to be even more valuable with the addition of further discussions that may engage readers interests.

Response to Reviewer 2

We have thoroughly reviewed and addressed all the comments and suggestions provided by the reviewer. Specifically:

  1. Spelling and Formatting Corrections: We have rechecked the entire manuscript for any spelling and formatting errors and made the necessary corrections. The issues in line 19 of the Introduction and the incorrect references on lines 35-36 have been resolved.
  2. Improving Readability: In response to the suggestion to enhance the readability of the sections on ROS mechanisms and specific pathways, we have introduced subheadings. These subheadings are designed to break down the dense content into more digestible parts, making it easier for readers to follow the discussion.
  3. Deepening Clinical Relevance: We have expanded the discussion to include more clinical data, specifically examining how ROS levels correlate with cancer prognosis, treatment responses, and survival rates. This addition bridges the gap between the molecular mechanisms of ROS and their implications in actual patient outcomes.
  4. Exploration of Additional Topics: We have incorporated a discussion on the current use of antioxidant medications or supplements in cancer therapy, how ROS levels vary across different stages of cancer, and the role of genetic modifiers in ROS sensitivity. This includes an exploration of how these factors contribute to the development of ROS-targeting therapeutic agents.

However, we respectfully note that the review is a survey of pathways and mechanisms pertaining to the effects of ROS on PPIs and cannot cover all aspects of the roles of ROS in different types of cancer and in the role of interventions. This is a separate and interesting topic that has to be addressed in a different review. Nevertheless, we included a short introduction on these aspects because we are in agreement with the reviewer that they are significant.

Reviewer 2 Report

Comments and Suggestions for Authors

This is an important issue and a well-organized review manuscript about "ROS and cancer." However, the manuscript has some minor spelling or miss-writing errors. Please re-check it all. 

1. line 19, Introduction

2. line 35-36, ref [ 10, 10? ...] [13,13?, ..]

Aside from some visible spelling and formatting mistakes, I suggest the following regarding the content:

The sections on ROS mechanisms and specific pathways are particularly dense. I suggest including subheadings to significantly improve readability.

There is a valuable opportunity to deepen the discussion of clinical relevance. I recommend integrating more clinical data to bridge the gap between the molecular mechanisms of ROS and actual patient outcomes. For example, how ROS levels correlate with cancer prognosis, treatment responses, or survival rates.

Additionally, you may consider exploring the following topics: 1. Current use of

antioxidants medication or supplements in cancer therapy. 2. how ROS levels vary across different stages of cancer 3. the role of genetic modifiers in ROS sensitivity and the development of ROS-targeting therapeutic agents.

Currently, the article offers a thorough overview of mechanistic details, and it has the potential to be even more valuable with the addition of further discussions that may engage readers interests.

Comments on the Quality of English Language

There are some minor spelling errors or miss-writing errors in the manuscript. Please re-check it all. 

Author Response

(The authors gave the same response as above.)
